# Mapping of quantitative trait loci for traits linked to fusarium head blight in barley

**Piotr Ogrodowicz[1], Anetta Kuczyńska[1]\*, Krzysztof Mikołajczak[1], Tadeusz Adamski[1], Maria Surma[1], Paweł Krajewski[1], Hanna Ćwiek-Kupczyńska[1], Michał Kempa[1], Michał Rokicki[2], Dorota Jasińska[2]**

**1** Institute of Plant Genetics, Academy of Sciences, Poznan, Poland, **2** Poznan Plant Breeding Station, Kasztanowa, Tulce, Poland

\* akuc@igr.poznan.pl

**Data Availability Statement:** All relevant data are within the paper and its Supporting Information files.

## Abstract

Fusarium head blight (FHB) is a devastating disease occurring in small grain cereals worldwide. The disease results in the reduction of grain yield, and mycotoxins accumulated in grain are also harmful to both humans and animals. It has been reported that response to pathogen infection may be associated with the morphological and developmental traits of the host plant, e.g. earliness and plant height. Despite many studies, effective markers for selection of barley genotypes with increased resistance to FHB have not been developed. In the present study, we investigated 100 recombinant inbred lines (RIL) of spring barley. Plants were examined in field conditions (three locations) in a completely randomized design with three replications. Barley genotypes were artificially infected with spores of *Fusarium culmorum* before heading. Apart from the main phenotypic traits (plant height, spike characteristic, grain yield), infected kernels were visually scored and the content of deoxynivalenol (DON) mycotoxin was investigated. A set of 70 Quantitative Trait Loci (QTLs) were detected through phenotyping of the mapping population in field conditions and genotyping using a barley Ilumina 9K iSelect platform. Six loci were detected for the FHB index on chromosomes 2H, 3H, 5H, and 7H. A region on the short arm of chromosome 2H was detected in which many QTLs associated with FHB- and yield-related traits were found. This study confirms that agromorphological traits are tightly related to FHB and should be taken into consideration when breeding barley plants for FHB resistance.

## Introduction

Fusarium head blight (FHB) or scabs affects different species of crops around the world. The infection is caused by several fungal pathogens, including *Fusarium culmorum* (W. G. Sm.) *Sacc* and *Fusarium graminearum* (teleomorph stage: *Gibberella zeae*). *Fusarium culmorum* has been found to dominate in regions with warm and humid conditions, whereas *Fusarium graminearum* has been associated with cool, wet, and humid conditions [1]. *Fusarium* spp. produce trichothecene—deoxynivalenol (DON) [2]. This mycotoxin disrupts normal cell function by inhibiting protein synthesis [3], which results in reduced grain quality and yield

**Funding:** This work was financially supported by Polish Ministry of Agriculture and Rural Development (grant no HOR hn-501-19/15 Task 88).

**Competing interests:** The authors have declared that no competing interests exist.

performance. Floret sterility and deformed kernels contribute to significant yield loss [4]. In Europe, 15–55% of barley products are contaminated with DON [5].

DON poses a genuine threat to human and livestock health. This mycotoxin is also known as "vomitoxin" due to its emetic effects after consumption [6]. DON levels present in barley (*Hordeum vulgare* L.) and wheat (*Triticum aestivum* L.) infected with FHB vary according to the time of infection and environmental factors. It is well known that infection is favored by moist and warm conditions [7, 8]. While the presence of scab can be determined through visual inspection, the presence of DON cannot. Assessment of disease severity is based on the ratio of symptomatic spikelets on each spike and the proportion of infected spikes in tested plants [9]. Although this method is widely used in the screening of resistant germplasms, the results are subjective. For identification and quantification of mycotoxins in barley grain different types of chromatography are commonly used [10, 11]. However, due to the time-consuming and costly nature of these methods, commercial immunometric assays, such as enzyme-linked immunosorbent assay (ELISA), are frequently applied for monitoring of mycotoxin content [12, 13].

Disease control is achieved by the deployment of resistant cultivars. However, breeding for FHB resistance has proven to be difficult due to the complex inheritance of resistance genes [14] and the strong genotype-by-environment interaction [15].

One of the several crop species most vulnerable to FHB infection is barley (*Hordeum vulgare* L.). This species is a cereal crop of major importance, and is ranked as the fourth grain crop worldwide in terms of production volume [16]. Its major uses include both animal feed and as a component of human nutrition [17, 18]. In addition, barley is a model plant in genetic studies due to colinearity and synteny across rye, barley, and wheat genomes [19].

Fusarium poses a tangible threat for barley plants, especially in regions that are prone to periods of wet weather during the flowering stage [4]. Host plants are most vulnerable to infection during anthesis due to development of fungal spores on anthers and pollen containing nutrients [20]. Numerous morphological traits have been shown to be associated with FHB resistance in barley [21], and in this regard heading date, plant height, and spike traits (linked to spike compactness) are mostly investigated [22, 23]. Days to heading is often negatively correlated with FHB susceptibility and usually results in disease escape [24]. Hence, using the least susceptible varieties with different flowering dates may reduce the risk of FHB. Two categories of resistance to FHB are generally recognized: type I (resistance to initial infection) and type II (resistance to fungal spread within the spike) [25]. Another type of resistance has been described as a third type and is related to accumulation of mycotoxins within the grain [26].

Studies designed to determine the number and chromosomal location of loci contributing to FHB resistance and the accumulation of DON are urgently needed for resistance breeding efforts. It is known that resistance to FHB is a complex trait controlled by multiple genes and affected by several environmental factors [27, 28]. Quantitative trait loci (QTLs) have been identified for quantitative disease resistance in wheat and barley [4]. Furthermore, resistance to both FHB and DON levels have been mapped to all seven barley chromosomes [29, 30], and the most common regions related to FHB resistance have been previously reported to be located on chromosomes 2H and 6H [3, 25, 31]. Other traits, including awned/awnless ears [26] and spike compactness [32], have also been studied. Plant height is another frequently investigated parameter, and a negative correlation of this trait with type I FHB susceptibility has been frequently documented [33].

Molecular markers have become increasingly important for plant genome analysis, and different classes of DNA markers have been developed and implemented over time [34]. A new genotyping platform was introduced in 2009 that contained larger numbers of markers based on SNP discovery from Next Generation Sequencing data using the oligo pool assay from

Illumina as a marker platform [35] to improve the genotyping process. Based on these analyses, the 9K iSelect chip was developed which contains 7864 SNPs [36] and enables genotyping with high efficiency and reduced costs. Indeed, in the current study, this chip was employed due to the favorable tradeoff between genotyping costs and marker density.

The overall aim of the present study was to map quantitative trait loci for agronomic properties in a biparental population grown in field conditions and subjected to artificial infection with *Fusarium*. Evaluation of disease severity was based on visual assessment of infection and content of deoxynivalenol.

## Materials and methods

### Plant material

A 100-RIL population of spring barley (hereafter referred to as LCam) obtained from the cross between the Polish cultivar Lubuski and a Syrian breeding line—Cam/B1/CI08887//CI05761 (hereafter referred to as CamB) was studied in field conditions, together with both parental forms. The plant materials were described in detail in Ogrodowicz et al. [37] and parental genotypes were chosen on the basis of earlier studies conducted by Górny et al. [38].

### Field experiments

Experimental fields belonging to the Poznan Plant Breeding Company (PPB) in three locations were used for the present studies: Nagradowice (NAD–Western Poland, 52°19′14″N, 17°08′54″E), Tulce (TUL—Western Poland, 52°20′35.2″N 17°04′32.8″E), and Leszno (LES—Western Poland, 51°50′45″N 16°34′50″E). At each location, experiments were performed in randomized blocks with three replications. The effects of *Fusarium* infection were evaluated during the 2016 growing season. The two experimental variants consisted of un-inoculated (control) and inoculated plants. Seeds were sown on 1 m$^2$ plots. Control rows were established at a distance of 20.0 m from the plots designated for inoculation. This isolation was necessary to protect plants against infection during inoculation.

### Inoculum preparation

*Fusarium culmorum* isolates were incubated on wheat grain (50 g) in 300 ml Erlenmeyer glass flasks for five weeks. The colonies were covered with 15 ml of sterile distilled water. Inoculum was prepared just before the inoculations by liquid cultures of *Fusarium culmorum* (isolate KF846) and 0.0125% TWEEN®20 (Sigma-Aldrich Chemie GmbH). Inoculum concentration was adjusted to 105 spore/ml. Inoculation was performed at the flowering stage (Zadoks scale 65). Mist irrigation to promote fungal infection was performed for three days in the field using a sprinkler system with DN881A-type sprinkler heads equipped with 1.50-mm-diameter nozzles (Sun Hope Inc., Meguro-ku). Water was applied three times daily (at 07.00, 13.00, and 19.00) for 15 min at each interval.

### Agronomic traits

Agronomic traits were classified into three categories: traits associated with spike characteristics [number of spikelets (NSS), number of kernels (NGS), length of spike without awns (LS), numbers of sterile spikelets per spike (sterility), spike density (density), grain traits (grain weight per spike (GWS), grain yield per plot (GY), average weight of 1000 grains (TGW)], heading day and plant height [heading date (HD) and length of main stem (LSt)]. The traits measured with ontology annotation are listed in Table 1.

**Table 1. List of phenotypic traits with description, abbreviations, measured units and ontology annotation.**

| Trait (unit) | Trait description | Abbrev. | Annotation |
|---|---|---|---|
| Number of spikelets per spike | Number of spikelets in spike from 10 randomly selected spikes in a plot | NSS | http://purl.obolibrary.org/obo/TO_0000456 |
| Number of grains per spike | Number of grains collected from 10 randomly selected spikes in a plot | NGS | http://purl.obolibrary.org/obo/TO_0002759 |
| Length of spike (cm) | Length of spike from 10 randomly selected spikes in a plot (without awns) | LS | http://purl.obolibrary.org/obo/TO_0000040 |
| Rate of sterile spikelets per spike | Fraction of sterile spikelets per spike, calculated as a ratio of number of spikelets per spike (NSS) to number of grains per spike (NGS) | Sterility | http://purl.obolibrary.org/obo/TO_0000436 |
| Spike density | Number of spikelets per unit length (centimeter) of spike calculated by dividing the number of spikelets per spike by the length of the spike | Density | http://purl.obolibrary.org/obo/TO_0020001 |
| Grain weight per spike (g) | Average weight of grain per spike, calculated from 10 randomly selected spikes in a plot | GWS | http://purl.obolibrary.org/obo/TO_0000589 |
| Grain yield (g) | Weight of grain harvested per plots | GY | http://purl.obolibrary.org/obo/TO_0000396 |
| 1000-grain weight (g) | Average weight of 1000 grains, calculated as 1000 * average weight of one grain for 10 spikes in a plot | TGW | http://purl.obolibrary.org/obo/TO_0000382 |
| Heading date (number of days) | Number of days from beagining of year to emergence of inflorescence (spike) from the flag leaf(51 BBCH), assessed when spikes emerged on at least 50% of plants | HD | http://purl.obolibrary.org/obo/TO_0000137 |
| Length of main stem (cm) | Average of measurements of length of stem from ground level to the end of spike (without awns) for 10 randomly selected plans in a plot | LSt | http://purl.obolibrary.org/obo/TO_0000576 |
| FHB index (%) | Spike infection, calculated as (the percentage of spikelets affected within a spike * the percentage of infected spikes per plot)/100 | FHBi | http://purl.obolibrary.org/obo/TO_0000662 |
| DON concentration (ppb) | Deoxynivalenol content of the grain | DON | http://purl.obolibrary.org/obo/TO_0000669 |
| Number of damaged kernels | Number of kernels classified as damaged (pinkish or discoloured) per 10 randomly selected spikes per plot | FDKn | |
| Weight of damaged kernels (g) | Weight of kernels classified as damaged (pinkish or discolored) per 10 randomly selected spikes per plot | FDKw | |
| Number of healthy kernels | Number of kernels classified as healthy per 10 randomly selected spikes per plot | HLKn | |
| Weight of healthy kernels (g) | Weight of kernels classified as healthy per 10 randomly selected spikes per plot | HLKw | |

## Evaluation of disease symptoms

Disease development was visually scored (Table 1) using the Fusarium Head Blight index (FHBi) computed as (percentage of infected spikelets within a spikes * percentage of infected spikes per plot)/100. The assessments were performed 20 days after inoculation. After harvest, Fusarium-damaged kernels (FDK) were observed as the number (FDKn) and weight (FDKw) of kernels—classified as pinkish or discolored (S1 and S2 Figs). Kernels that appeared to be healthy were scored as healthy-looking kernels (HLKn and HLKw). The FDK and HLK rates were estimated for infected and controlled kernels at one location (NAD). DON content (ppm) from infected grain samples (in each experiment with three replications) was assessed using a Ridascreen®DON competitive enzyme immunoassay kit (R-Biopharm AG, Darmstadt, Germany) according to the manufacturer's instructions. For the DON assay, 5 g samples of kernel were ground and 100 ml of distilled water was added. Samples were shaken vigorously for three minutes (manually). After incubation, samples were filtered through Whatman No. 1 filters; 50 μl of the filtrate per well was used in the test. Absorbance was measured at 450 nm with a spectrophotometer (Chromate Microplate Reader), and data were evaluated with RIDA®SOFT Win software. Within a single location (NAD, TUL, LES), samples obtained from plants grown under controlled conditions (exposed to natural infection) were pooled together as one sample and assayed as above. For each sample, three repetitions (biological

replicates) were performed (three repetitions for each inoculated condition and three repetitions for one representative controlled condition).

## Genotyping

Genomic DNA was extracted from young leaf tissue as described in Mikołajczak et al. 2016 [39]. DNA quantity and concentration were measured with a NanoDrop 2000 spectrophotometer (Thermo Scientific™). DNA samples were diluted to ~ 50 ng/μL and sent to Trait Genetics, Gatersleben, Germany (http://www.traitgenetics.com) for genotyping using the barley iSelect SNP chip. This chip contains 7,842 SNPs that comprise 2,832 of the existing barley oligonucleotide pooled assay (BOPA1 and BOPA2) SNPs discovered and mapped previously [40, 41], as well as 5,010 new SNPs developed from Next Generation Sequencing data [36, 42]. SNPs which were not polymorphic between the parents, contained more than 10% of missing values, or with minor allele frequency < 15% were removed from the data set.

## Linkage map

Genetic map was calculated using JoinMap 4.1 software [43]. All markers were analyzed for goodness of fit using a chi-square test with α = 0.05. A segregation ratio of 1:1 was expected. Markers with other segregation ratios were categorized as markers with segregation distortion. The localization of markers was designated using the maximum likelihood algorithm. Markers were assigned to linkage groups by applying the independence LOD (logarithm of the odds) parameter with LOD threshold values ranging from 6.0 to 9.0. The recombination frequency threshold was set at level 0.4. Recombination fractions were converted to map distances in centimorgans (cM) using the Kosambi mapping function. A map was drawn using MapChart 2.2.

## Data analysis and QTL mapping

Observations for RILs were processed by analysis of variance in a mixed model with fixed effects for location, treatment, and location × treatment interaction, and with random effects for line and interaction of line with location and interaction of line with location and treatment. The residual maximum likelihood algorithm was used to estimate variance components for random effects and the F-statistic was computed to assess the significance of the fixed effects. Pearson correlation coefficients between all the analyzed traits were calculated. QTL analysis was performed for the linkage map with the mixed model approach described by Malosetti et al. [44], including optimal genetic correlation structure selection and significance threshold estimation. The threshold for the–log10($P$-value) statistic was computed using the method of Li and Ji [45] to ensure that the genome-wide error rate was < 0.01. Selection of the set of QTL effects in the final model was performed at $P < 0.05$; $P$-values for the Wald test were computed as the mean from the values obtained by adding and dropping the QTL main and interaction effects in the model. All the above computations were performed with Genstat 18 [46]. RILs with a fraction of missing genotypic data smaller than 20% were used to map QTL. QTL identification was performed for all traits.

The detected QTLs were labeled using a system described for wheat and *Arabidopsis* [47, 48], with minor modifications. The QTLs names consist of the prefix Q followed by a two- or three-letter descriptor of the phenotype (abbreviation of the trait name), an indicator for the laboratory, the chromosome number, and a serial number. For traits linked to FDK and HLK, the QTL names were extended by adding the letter "w" or "n" for loci found for trait weight of FDK, HLK, and number of FDK and HLK, respectively.

QTL effects in individual trials were considered major if the fraction of explained variance exceeded 12.32% (upper quartile of the distribution of explained variance) according to the rules employed by [49] and [50] (with minor modifications).

The barleymap pipeline (http://floresta.eead.csic.es/barleymap) [51] was used to identify SNP positions in the reference Morex genome and gene annotations linked to potential candidate genes located in the vicinity (intervals around markers extended by ±2cM) of the particularly robust QTL. Overrepresentation analysis of Gene Ontology (GO) terms in QTL regions was performed using the GO annotation of barley genes downloaded from Ensembl Plants Genes (rel. 43) database in Bingo 3.0.3 [52] (hypergeometric test, Benjamini-Hochberg [53] FDR corrected p-values < 0.05).

## Results

### Phenotypic analysis

The parents of the LCam population were characterized with 10 agronomical traits under two different conditions (infection and control treatments). Evaluation of disease severity was studied by using measurements of six FHB-related traits in both previously mentioned conditions. The distributions of trait values among RILs are visualized in Fig 1. Raw data are available at data repository Ćwiek-Kupczyńska et al. [54].

Lubuski showed higher mean values of traits linked to yield performance (e.g. GWS, GY) (S1 Table) than CamB. CamB showed a lower mean value of HD in all trials and under both types of treatments (heading for CamB was 11 days earlier than for Lubuski). A substantial GY decline was observed for Lubuski in conditions of infection (40.1%). In comparison, for CamB a lower relative decline for GY was observed (17.3%). Both mean values for FHBi and traits associated with visual evaluation of *Fusarium* symptoms (FDK) increased during infection. For CamB, a higher mean value of DON concentration was noted in comparison to that of Lubuski. For both parental forms low concentrations of mycotoxin were also observed in control conditions than under infection.

The mean values of the studied traits for RILs are presented in S2 Table. Relatively high values of variation coefficients were observed in the NAD location under infection for the following traits: NSS, NGS, Density, GWS, and TGW. In the LES location, very high values of CV were noted for traits FHBi and DON under control conditions. FHBi varied across locations with the mean value ranging from 1.89 to 2.26 under infection and from 0.62 to 0.99 in the control conditions (S2 Table). The amount of DON, measured in grains from infected plants, varied from 8.06 ppm (NAD) to 39.99 ppm (TUL). Mean DON values of 26.43, 25.68, and 27.14 ppm for infection at LES, NAD, and TUL were observed, respectively. In control conditions, relatively high coefficients of variation were noted for DON and FHBi.

Analysis of variance indicated significant effects of location and treatment for all traits (P<0.001) with some exceptions (S3 Table). In all cases, variance components for all types of interactions were smaller than those for lines. For FHBi, a significant line × location interaction was noted, while no signicant interaction was observed for line × treatment in this case. An insigificant effect was noticed in terms of the interaction line × location for DON content. The values of correlation coefficients between the studied traits and FHBi were generally low (Fig 2). FHBi was negatively correlated with all studied traits (exeptions: FDKw and Sterility). In LES location positive significant correlation between FHBi and Density was recorded, whereas negative correlation coefficients were noted between these traits in other two locations. Positive correlation was recorded between DON content and FHBi in one of three locations (TUL) for both type of treatments. No significant correlations between DON content and other agronomic traits were observed.

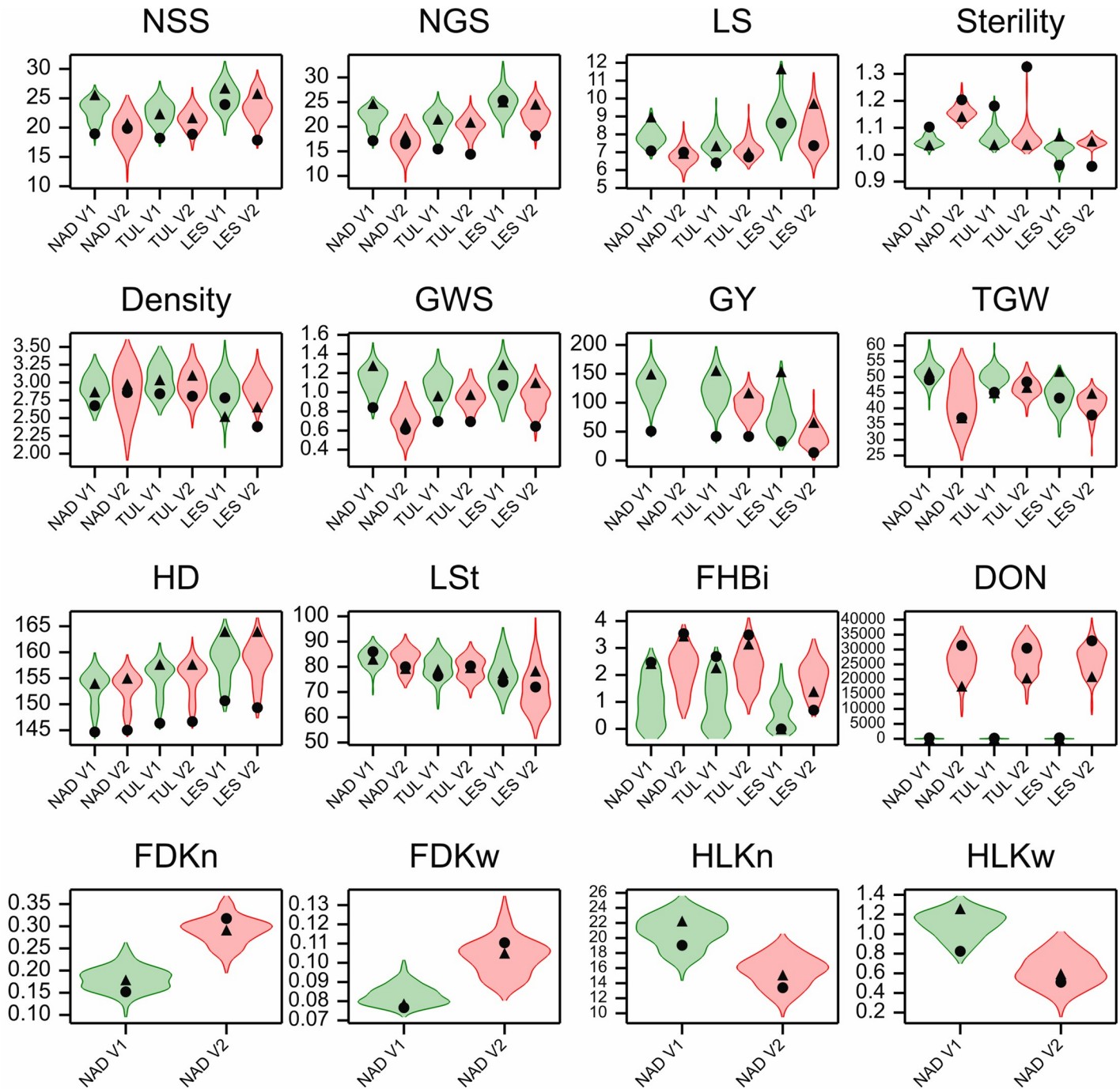

**Fig 1. Violin plots for traits measured in the LCam population in control (V1, green) and infected (V2, red) conditions in three locations.** Black symbols: triangle, Lubuski; dot, CamB.

## Linkage map construction

The constructed genetic map comprised 1947 SNPs distributed in seven linkage groups. The map length was 1678 cM with an average marker interval of 0.86 cM. The shortest chromosome was 6H, which harbored 250 markers with a genetic length of 141 cM and an average

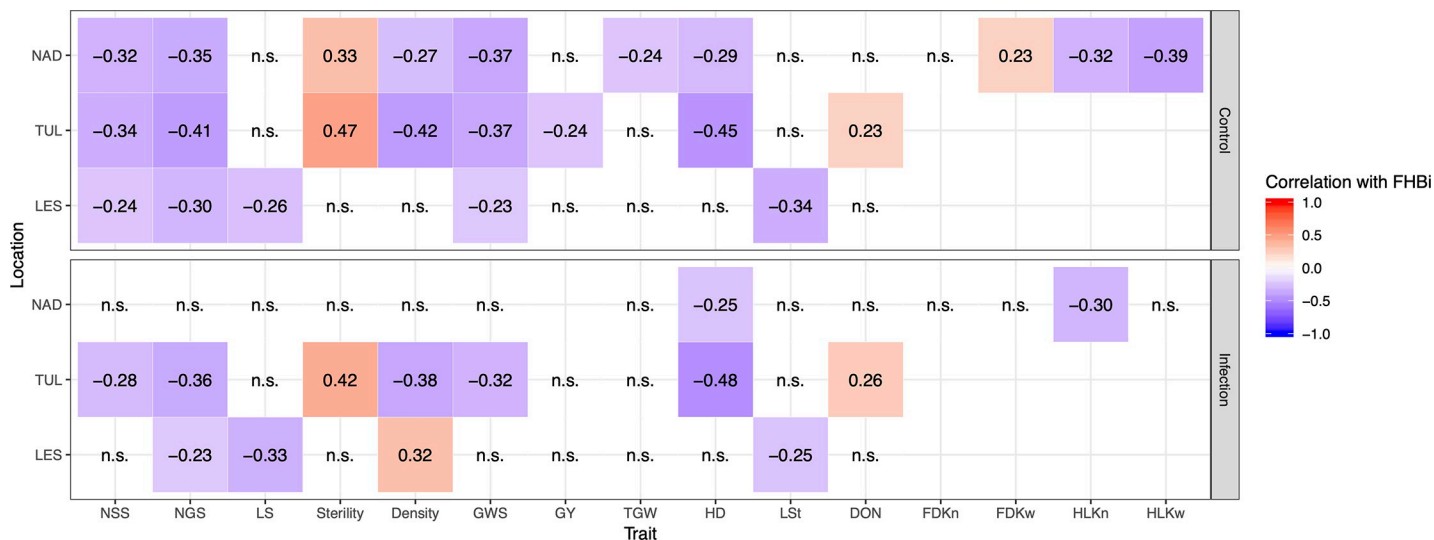

**Fig 2. Correlation coefficients between FHBi and studied traits recorded under two treatments at three locations (n.s.- not significant; correlations shown are significant at the P < 0.01 level).**

interloci distance of 0.56 cM. The longest chromosome was 2H, harboring 368 markers with a genetic length of 291 cM and an average interloci distance of 0.79 cM. The number of markers, marker density, and map length for individual chromosomes are listed in Table 2.

## QTL analysis

A total of 70 QTLs for all studied traits were found for the LCam population (S3, S4 and S5 Figs). The numbers of QTLs were 7, 24, 5, 6, 17, 4, and 7 for chromosomes 1H, 2H, 3H, 4H, 5H, 6H, and 7H, respectively. Moreover, 46 QTLs presented major effects and 38 presented a QTL × E interaction. The largest number of QTLs was detected for NSS and TGW (eight QTLs were identified for each trait), and the smallest for FDK (two QTLs were detected for FDKn and FDKw). Fourteen QTLs were classified as major loci and 56 QTLs were described as minor loci. Detailed information, including location, peak marker, additive effects, and explained phenotypic variance for each QTL and trait is presented in S4 Table.

**Spike characteristics.** For the number of spikelets per spike, eight QTLs were detected in chromosomes 1H, 2H, 3H, and 5H. The major QTL (*QNSS.IPG-2H_1*) on chromosome 2H (SNP marker *BK_12*) showed the most significant effect for this trait and explained a large proportion of phenotypic variance (4.79–71.81%). In this case, significant QTL × E interaction was noted and Lubuski alleles conferred a positive effect in increasing this trait. A second locus positioned at 98.67 cM on chromosome 5H also showed a highly significant association with

**Table 2. Map details across each chromosome.**

| Characteristic | Chromosome | | | | | | | Total |
|---|---|---|---|---|---|---|---|---|
| | 1H | 2H | 3H | 4H | 5H | 6H | 7H | |
| Number of mapped markers | 156 | 368 | 324 | 329 | 324 | 250 | 196 | 1947 |
| Number of loci | 3 | 13 | 5 | 5 | 13 | 4 | 6 | 49 |
| Map length (cM) | 232 | 291 | 241 | 215 | 295 | 141 | 263 | 1678 |
| Mean distance between markers (cM) | 1.48 | 0.79 | 0.74 | 0.65 | 0.91 | 0.56 | 1.30 | 0.86 |
| Number of distorted markers (%) | 11.1 | 9.4 | 13.1 | 15.8 | 6.2 | 11.7 | 8.5 | |

NSS (LogP statistics = 16.95). Chromosome 1H was the location of the last major QTL (*QNSS. IPG-1H_1*) in the vicinity of marker *BOPA1_4625–1413*. The remaining five NSS QTLs showed minor effects. Out of the eight QTLs detected for NSS, three (*QNSS.IPG-1H_2*, *QNSS. IPG-2H_2* and *QNSS.IPG-3H_2*) were associated with a significant increase in this trait contributed by CamB alleles.

Five QTLs were found for the number of grains per spike. *QNGS.IPG-2H* was found in the vicinity of marker *BK_12*. This locus was positioned at 22 cM on chromosome 2H. The second major QTL was detected on chromosome 5H in the vicinity of marker *BOPA2_12_30929*. For this QTL, no significant additive effects were recorded in the NAD location. The other QTLs (*QNGS.IPG-1H_1*, *QNGS.IPG-1H_2* and *QNGS.IPG-5H_2*) were classified as minor QTLs. All QTLs for NGS were with alleles of Lubuski, contributing to an increasing of number of grains per spike.

Three QTLs were reported for the length of spike (*QLS.IPG-1H*, *QLS.IPG-2H*, and *QLS. IPG-5H*). All detected loci were classified as major (≥12.32% PVE) and the effects of these QTLs were stable over environments (treatments). All QTLs were associated wih a significant increase in LS contributed by Lubuski. The main QTL was found on chromosome 2H in the vicinity of marker *BK_13*. In total, five QTLs were identified for sterility. On chromosome 2H, two major QTLs were detected (*QSte.IPG-2H_1* and *QSte.IPG-2H_2*). The first, *QSte.IPG-2H_1*, was located in the vicinity of marker *SCRI_RS_154030* and showed the highest LogP value of all detected QTLs controlling this trait. The second sterility QTL was located on chromosome 2H 5.6 cM from marker *SCRI_RS_230497*. One major QTL (*QSte.IPG-5H_2*) was detected on chromosome 5H. None of the mentioned QTLs had significant additive effects in the control condition at the LES location or in the infection condition at the NAD location. On chromosome 7H, a minor QTL for sterility was identified, namely *QSte. IPG-7H*. All QTLs detected for this character were with alleles of CamB contributing to the increase in sterility with the exception of *QSte.IPG-5H_1*, where Lubuski alleles determined the increase. Interaction with the environment was found for all but one detected QTL (an exeption was *QSte.IPG-7H*).

Six QTLs controlling density were detected on chromosomes 2H and 5H with a PVE ranging from 0.01 to 31.43%. Half of those QTLs displayed significant QTL × E interaction. The major QTL (LogP = 15.17) was found on the upper arm of chromosome 2H mapped in marker *BK_22*. Concurrently, this QTL was the only locus associated with Density, where Lubuski alleles conferred a positive effect in increasing this trait, while CamB alleles at the other five QTLs contributed positively to Density. The second major QTL (*QDen.IPG-2H_2*) was also found on chromosome 2H at position 113.9 cM. On chromosome 2H two other minor QTLs were identified for Density QTL (*QDen.IPG-2H_2* and *QDen.IPG -2H_4*) with a stable effect, mapped in the vicinity of *BOPA1_5537–283*. *QDen.IPG -5H_1* was also found on chromosome 5H at position 93.9 cM. The additive effects of this QTL were significant at only two locations (NAD and TUL). For Density, two minor QTLs were found, *QDen.IPG-2H_3* and *QDen.IPG-5H_2*, for which the smallest LogP values were recorded for Density.

**Grain traits.** Grain weight per spike was mapped to seven loci. The major GWS QTL (*QGWS.IPG-2H_1*) was found on chromosome 2H in the vicinity of marker *BK_22*. This locus, with a PVE ranging from 31.74–57.92%, was the only GWS QTL where no significant QTL × E interaction was detected. *QGWS.IPG-5H* was found on chromosome 5H and the nearest marker (*BOPA1_4795–782*) was 1.37 cM away from the corresponding QTL peak. Two other major QTLs (*QGWS.IPG-7H_1* and *QGWS.IPG-7H_2*) controlling GWS were reported on chromosome 7H. Both of these QTLs had a significant additive effect at only in single location. A minor *QGWS.IPG-4H_2* was found on chromosome 4H at position 127.40 cM. Lubuski contributed to the increase in GWS for all detected QTLs for this trait (except for

two QTLs; minor *QGWS.IPG -2H_2* and major *QGWS.IPG-4H_1* on chromosomes 2H and 4H, respectively).

Of the four QTLs found for grain yield, only one was classified as major (PVE > 12.32%). In additon, no significant additive effects were noticed for any loci in infection conditions for the NAD location. The major *QGY.IPG-2H* was located on chromosome 2H and linked to marker *BK_22*. No QTL × E interaction was found for GWS QTLs detected in the mapping population and in all cases positive alleles were attributed to Lubuski.

Eight QTLs were reported for thousand grain weight. *QTGW.IPG-2H_1* and *QTGW.IPG-4H_1* were identified on chromosomes 2H and 4H, respectively, but their additive effects were significant only in infection (LES) and control conditions (NAD). On chromosome 4H, TGW QTL was found with a stable and positive effect from the Lubuski genotype. *QTGW.IPG-6H_2* locus on chromosome 6H was determined by CamB alleles contributing positively to TGW. In this locus, no significant QTL × environment interaction for TGW was observed. Major *QTGW.IPG-7H_1* with stable effects from the CamB allele significantly increasing TGW was identified on chromosome 7H. On the same chromosome *QTGW.IPG-7H_2* was found, but the additive effects of this QTL were significant in only three treatments. *QTGW.IPG-2H_2* and *QTGW.IPG-6H_1*, detected on chromosome 2H and 6H, respectively, were classified as minor QTLs.

**Heading day and height.**   Two QTLs (*QHD.IPG-2H* and *QHD.IPG-5H*) were reported for heading date. The major QTL was located on chromosome 2H in the vicinity of marker *BK_22*. The "late" allele (high HD value) was contributed by Lubuski. In contrast, at the second locus, classified as a minor QTL, the CamB alleles conferred a positive effect by increasing this trait. For both loci, no QTL × E interaction was detected.

Seven loci for length of the main stem were found in the LCam population. The major locus (*QLSt.IPG-2H_1*) was detected on chromosome 2H in the vicinity of marker *BK_13* at position 21 cM. This QTL explained a large portion of the variance for LSt (from 13.88 to 41.68%). The Lubuski alleles contributed to the increase in LSt at this locus. The second major QTL was reported on chromosome 1H with stable and positive effects on the length of the main stem contributed by Lubuski. *QLSt.IPG-4H_2* and *QLSt.IPG-5H* were identified on chromosomes 4H and 5H, respectively. These QTLs were classified as major loci, but their additive effects were not significant in some treatments (e.g. control conditions in NAD location). Three minor LSt loci were found: *QLSt.IPG-2H_2*, *QLSt.IPG-3H*, and *QLSt.IPG-4H_1* on chromosomes 2H, 3H and 4H, respectively.

**Fusarium symptoms and DON content.**   Six QTLs were reported for the FHB index. The major QTL (*QFHBi.IPG-2H_1*) was found on chromosome 2H in the vicinity of marker *BOPA1_5880–2547* at position 23.10 cM. The CamB alleles positively contributed to the increase in the FHB index at this locus and a significant QTL × E interaction was detected for *QFHBi.IPG-2H_1*. On the same chromosome, another FHBi QTL was reported which was located at position 87.70 cM, but additive effects of this locus were significant in only one location (TUL). The next major locus (*QFHBi.IPG-2H_3*) was also detected on chromosome 2H with stable and positive effects of CamB alleles responsible for increasing the FHBi. In contrast, the EuropLubuski alleles conferred a positive effect in increasing the FHBi at the locus found on chromosome 5H (*QFHBi.IPG-5H*). No significant additive effects were detected in the LES location for this QTL. Two minor loci – *QFHBi.IPG-3H* and *QFHBi.IPG-7H* – were found on chromosomes 3H and 7H, respectively.

Four QTLs were found for traits linked to *Fusarium*-damaged kernels. These loci were located on chromosomes 5H and 6H. The major *QFDKn.IPG-5H* was detected in the vicinity of *SCRI_RS_165578*, where Lubuski genotype significantly increased the FDKn. The second major locus (*QFDKw.IPG-5H*) was identified at position 87.80 cM and showed positive effects

on this trait contributed by Lubuski alleles. Two remaining loci on chromosome 6H (*QFDKn.IPG-6H* and *QFDKw.IPG-6H*) were classified as minor QTLs.

Five QTLs were detected for traits associated with HLKw and HLKn. The major *QHLKn.IPG-2H_2* was found on the short arm of chromosome 2H (marker *BK_13*) and showed stable and positive effects of Lubuski genotype alleles which contributed to the increase in HLKn. Two minor QTLs were recorded for HLKn on chromosomes 2H and 5H. No significant QTL × E interaction was detected for either locus. Two major loci (*QHLKw.IPG-2H* and *Q_HLKw.IPG-7H*) were found for the trait HLKw. The Lubuski alleles were responsible for increasing HLKw in both loci, but only one QTL (*QHLKw.IPG-2H*) had stable effects.

In this study, no QTL for DON content was detected.

## Co-localized or pleiotropic QTLs

All QTLs linked to FHB on chromosomes 2H and 5H co-localized with other agronomic traits. A total of eight chromosomal regions (named A-G) harboring QTLs for the studied traits were defined. These regions (hotspots), listed in Table 3, were designed based on inter-QTL distances smaller than 2 cM. Five QTLs were reported in region A located on the long arm of chromosome 1H, and associated with SNP *BOPA1_4625–1413*. Region B identified on the short arm of chromosome 2H contained 10 loci. In most cases, QTLs from region B were detected in the vicinity of marker *BK_12* (S4 Fig). Out of the five QTLs detected in region C assigned to the same chromosome, four were found in the vicinity of marker *BOPA2_12_109 37*. Region D harbored two QTLs found at the same position (127.40 cM). Region E (chromosome 5H) harbored six QTLs; *QDen.IPG-5H_1* and *QFHB.IPG-5H* were found in this region in the vicinity of marker *SCRI_RS_184066* and both *QNGS.IPG-5H_1* and *QLSt.IPG-5H* were detected in the vicinity of marker *BOPA2_12_30929*. On the same chromosome, the next region was noted (named region F). Of the four QTLs reported on this region, two were found in the vicinity of marker *SCRI_RS_206867*. Region G on chromosome 7H harbored two loci associated with marker *SCRI_RS_159555*.

QHLKw.IPG-2H was found in the vicinity to marker BK_12. In the same position a set of QTLs linked to different agronomic traits was found (Density, GWS, GY, HD, NGS and NSS —QDen.IPG-2H_1, QGWS.IPG-2H_1, QGY.IPG-2H, QHD.IPG-2H, QNGS.IPG-2H and QNSS.IPG-2H_1, respectively). QHLKn.IPG-2H_2, other QTL related to FHB, was detected in the vicinity of marker BK_13 –in the same position as QTLs related to LS (QLS.IPG-2H) and LSt (QLSt.IPG-2H_1). In both cases Lubuski contributed positively to the increase of the trait linked to HLK (HLKn and HLKw).

An overrepresentation analysis was performed (S5 Table) to identify enriched Gene Ontology (GO) terms–cellular component, molecular function, and biological process–associated with genes in three regions B, E, and F containing QTLs for FHB-related traits (Table 3). Genes annotated with overrepresented GO terms associated with FHB responses (glucuronosyltransferase activity, galactosylgalactosylxylosyl protein 3-beta-glucuronosyltransferase activity, transferase activity, transferring hexosyl groups) are listed in Table 4.

## Discussion

It is widely known that a mapping population derived from parents divergent in genetic composition allows high performance QTL analysis. In this study, RILs (named LCam) derived from a European variety (Lubuski) and a Syrian breeding line (CamB) were used for QTL analysis. Both parental forms were differentiated in terms of height, grain yield, HD, and resistance/tolerance to abiotic stress [37, 39]. CamB is unadapted to the Central European region and has undesired agonomical traits such as early heading and tall stature. Lubuski is an old

**Table 3. Regions harboring QTLs for traits with the names of the nearest SNP markers.**

| Name of region | Trait | QTL ID | Chromosome | Position (cM) | Nearest marker |
|---|---|---|---|---|---|
| A | NSS | *QNSS.IPG-1H_1* | 1H | 0.00 | BOPA1_4625–1413 |
| | NGS | *QNGS.IPG-1H_1* | 1H | 0.00 | BOPA1_4625–1413 |
| | LS | *QLS.IPG-1H* | 1H | 0.00 | BOPA1_4625–1413 |
| | LSt | *QLSt.IPG-1H* | 1H | 0,00 | BOPA1_4625–1413 |
| | GY | *QGY.IPG-1H* | 1H | 0.00 | BOPA1_4625–1413 |
| B | LS | *QLS.IPG-2H* | 2H | 21.00 | BK_13 |
| | LSt | *QLSt.IPG-2H_1* | 2H | 21.00 | BK_13 |
| | NSS | *QNSS.IPG-2H_1* | 2H | 22.00 | BK_12 |
| | NGS | *QNGS.IPG-2H* | 2H | 22.00 | BK_12 |
| | Density | *QDen.IPG-2H_1* | 2H | 22.00 | BK_12 |
| | GWS | *QGWS.IPG-2H_1* | 2H | 22.00 | BK_12 |
| | GY | *QGY.IPG-2H* | 2H | 22.00 | BK_12 |
| | HD | *QHD.IPG-2H* | 2H | 22.00 | BK_12 |
| | HLKw | *QHLKw.IPG-2H* | 2H | 22.00 | BK_12 |
| | FHBi | *QFHB.IPG-2H_1* | 2H | 23.10 | BOPA1_5880–2547 |
| C | Density | *QDen.IPG-2H_3* | 2H | 225.26 | BOPA2_12_10937 |
| | LSt | *QLSt.IPG-2H_2* | 2H | 225.26 | BOPA2_12_10937 |
| | GWS | *QGWS.IPG-2H_2* | 2H | 228.70 | BOPA2_12_10937 |
| | TGW | *QTGW.IPG-2H_2* | 2H | 228.70 | BOPA2_12_10937 |
| | NSS | *QNSS.IPG-2H_2* | 2H | 229,80 | SCRI_RS_174051 |
| D | TGW | *QTGW.IPG-4H_2* | 4H | 127.40 | BOPA1_2196–195 |
| | GWS | *QGWS.IPG-4H_2* | 4H | 127.40 | BOPA1_2196–195 |
| E | Density | *QDen.IPG-5H_1* | 5H | 93.90 | SCRI_RS_184066 |
| | FHBi | *QFHB.IPG-5H* | 5H | 95.60 | SCRI_RS_184066 |
| | NGS | *QNGS.IPG-5H_1* | 5H | 97.30 | BOPA2_12_30929 |
| | LSt | *QLSt.IPG-5H* | 5H | 97.30 | BOPA2_12_30929 |
| | LS | *QLS.IPG-5H* | 5H | 97.30 | BOPA2_12_30929 |
| | NSS | *QNSS.IPG-5H_1* | 5H | 98.67 | SCRI_RS_235055 |
| F | GY | *QGY.IPG-5H* | 5H | 285.20 | BOPA2_12_30533 |
| | NSS | *QNSS.IPG-5H_2* | 5H | 286.90 | SCRI_RS_206867 |
| | NGS | *QNGS.IPG-5H_2* | 5H | 286.90 | SCRI_RS_206867 |
| | HLKn | *QHLKn.IPG-5H* | 5H | 288.00 | SCRI_RS_165919 |
| G | GWS | *QGWS.IPG-7H_1* | 7H | 119.80 | SCRI_RS_159555 |
| | TGW | *QTGW.IPG-7H_2* | 7H | 119.80 | SCRI_RS_159555 |

cultivar with agro-morphological-physiological characters adapted to climatic conditions in Poland during a long cultivation period. With the aim of providing the genetic variability between the parents of the mapping populaion and increasing the chance of indentifying loci linked to FHB, we conducted field experiments using RILs derived from a cross between Lubuski and CamB genotypes.

FHB, caused by *Fusarium culmorum*, is a very important disease affecting crops on a global scale [9]. The pathogen is dominant in cooler areas like north, central and western Europe [55]. *F. graminearum* predominates in the warmer, humid areas of the world such as USA [56, 57]. Damage caused by *Fusarium* fungus includes reduced grain yield and grain functional quality, and results in the presence of the mycotoxin deoxynivalenol in FDK (even in grains without any visible symptoms). The development of FHB resistant crop cultivars is an important component of integrated breeding management [58, 59]. The objective of this investigation was to

**Table 4. Genes annotated with selected overrepresented GO terms from regions B, E and F.**

| Gene | Gene description | Position in region | | |
|------|------------------|-----|-----|-----|
| | | **B** | **E** | **F** |
| HORVU2Hr1G013590 | Glycosyltransferase | + | - | - |
| HORVU2Hr1G013630 | Glycosyltransferase | + | - | - |
| HORVU5Hr1G095010 | UDP-Glycosyltransferase superfamily protein | - | + | - |
| HORVU5Hr1G096240 | UDP-Glycosyltransferase superfamily protein | - | + | - |
| HORVU5Hr1G096260 | UDP-Glycosyltransferase superfamily protein | - | + | - |
| HORVU5Hr1G096310 | UDP-Glycosyltransferase superfamily protein | - | + | - |
| HORVU5Hr1G096320 | UDP-Glycosyltransferase superfamily protein | - | + | - |
| HORVU5Hr1G096340 | UDP-Glycosyltransferase superfamily protein | - | + | - |
| HORVU5Hr1G096360 | UDP-Glycosyltransferase superfamily protein | - | + | - |

identify QTLs for traits linked to yield performance in a recombinant inbred line population grown under a disease-free environment and under conditions of *Fusarium* infection.

FHB infection can be evaluated in different ways. In field conditions, FHB can be determined by visual inspection of the percentage of infected spikelets [60], and can be used to determine an FHB index [61]. After harvest, percentage of both FDK and HLK as described by the visual symptom score and weight of kernels were evaluated. In addition, DON concentration was quantified. In this study, Lubuski was less susceptible to FHB than CamB in all conditions in terms of DON accumulation. On the other hand, we observed a higher FHBi value for Lubuski plants during infection. This can be explained by the fact that symptomless grains may contain significant amounts of mycotoxins, while symptomatic grains within the same samples may not [62]. DON tests of grains harvested from the LCam population showed that some RILs showed lower DON content values than CamB, while other RILs were more susceptible than Lubuski. In all conditions, the mean values of agronomic traits were as expected, i.e. biotic stress conditions led to impaired yields. In control conditions, DON contamination represents the natural occurrence of FHB [63] and the level of mycotoxin accumulation varied significantly from those observed in LCam plants grown in conditions of infection.

Most of the correlation coefficients among the FHBi and other characteristics studied were negative and statistically significant (P<0.01). The Pearson correlation coefficient between FHBi and the two main traits of our interest, HD and LSt, was also significantly negative, which is in agreement with previous studies [64–66] where plants with lower FHB severities have usually been characterized by late heading and tall stature. Late-maturing plants may head during a time in the summer that is less suitable for infection, and tall plants avoid higher concentrations of inoculum near the surface of the soil [67]. Another study by Mesfin et al. [24] concluded that late HD may be linked to FHB resistance since the heads experience less exposure time to fungal spores. Moreover, a negative correlation coefficient was recorded between FHBi and Density, which can be explained by the fact that lax spikes dry faster and it is difficult for the pathogen to spread upward and downward on the spike [68].

Visual ratings for FHB in barley plants are usually conducted just before the spikes begin to lose chlorophyll, and thus disease symptoms can be easily scored. In some years, there are favorable conditions for *Fusarium* growth and DON accumulation throughout plant senescence, reducing correlations between FHB and DON because the FHB score does not accurately reflect the final disease level [69]. It is well known that symptomless spikes can be contaminated with DON [70]. Traditionally, mycotoxin determination has mainly been performed by chromatographic techniques [71, 72], although ELISA has been proposed as an alternative method to visual scoring and DON quantification for measuring FHB [73]. The

relationship between visual symptoms of FHB and DON content is highly variable ranging from none to a very strong positive relationship [69]. The difference in relationships may be due to differences among plant varieties, weather conditions, pathogen population and disease management practices [74]. As a consequence, FHB and DON values are not always closely correlated. In our study, positive significant correlation was found between FHBi and levels of DON only in one location.

Agronomic traits related to spike traits (e.g., spike density and sterility) have been reported to be linked to FHB resistance, but the association between traits and FHB vulnerability seems to be unclear. Steffenson et al. [75] reported that FHB severity was apparently higher in dense spike NILs than in lax spikes. A negative correlation between FHB severity and spike density was recorded in an experiment on a population derived from two-row and six-row barley plants [76]. In contrast, spike density had little or no effect in the study by Yoshida et al. [77] on barley NILs. Ma et al. [78] also reported an association between lax spike and the FHB reaction. Lax spikes may be related to FHB resistance due to their specific architecture that retains, presumably, less moisture within the whole spike (lax spike dry faster and it is difficult fo the pathogen to spread upward and downward of the spike). This decreases the pace of fungus spread [4]. Herein, negative correlations were detected between the traits Density and FHBi, indicating that spike compactness may be one of the factors enhancing FHB susceptibility. A positive correlation was also recorded between Sterility and FHBi in LCam plants, which means that FHB infection had negative effects on seed development, as expected.

The polymorphic SNP markers found in this study were distributed across all seven linkage groups in the LCam mapping population. Marker order and distances for SNPs generally matched previously published barley maps [40, 79]. The genetic map consisting of 1947 SNPs developed in this study, covering 1678 cM, is larger than other maps (e.g., that constructed by Wang et al. [80] covered 1375.8 cM).

Many bi-parental mapping studies have been carried out on barley to explain the genetic architecture of resistance to FHB and DON accumulation and to identify molecular markers that could be useful in breeding [24, 30, 81, 82]. FHB resistance has frequently been found to be associated with plant morphology parameters, and especially plant height, spike architecture, anther extrusion and HD. For this reason, the LCam population was also evaluated for HD, plant height, spike compactness, and other traits, which seem to be important from an agronomic point of view. Numerous QTL mapping studies in different crop species have revealed that QTLs associated with FHB resistance are coincident with QTLs linked to various agronomic and morphological traits [4, 24, 82]. Previous studies have used population sizes comparable to this study and successfully identified FHB QTL [24, 31, 83]. In our investigation, 70 QTLs were detected on seven barley chromosomes. A higher number of QTLs for agronomic traits was found on chromosome 2H, where the greatest number of FHB-linked QTLs was also identified.

In our study different tools for FHB evaluation have been used: among others: DON content estimation. No QTLs for DON content were detected but visual assessment of FHB severity like FHBi, FDK and HLK were employed here for evaluation of the level of FHB severity. In this study six, four and five QTLs were found for FHBi, FDK and HLK, respectively. The association between Fusarium head blight (FHB) intensity and DON accumulation in harvested grain is not fully understood. Varying degrees of association between Fusarium head blight intensity and DON accumulation in harvested grain have been reported in the literature, including situations with high positive correlations, low significant correlations, and negative correlations, as well as correlations close to zero [84–87]. Visual assessments of disease were usually made at Feekes GS 11.2, based on the proportion of the spike diseased, while DON was quantified in this study after harvest as the amount of DON per unit weight of a bulked sample

of ground kernel. The measurement of DON in an assay typically is a composite value for seeds with different levels of DON (including those with 0 ppm) and different levels of fungal colonization. In our study positive correlation between DON content and FHBi was observed only in one location which can be explained by the fact that the growth of the fungus and the production of DON are highly weather dependent [88, 89]. Moreover, DON concentration may have increased at differential rates in the different studies, affecting the relationship between DON sampled at harvest and disease assessed different developmental stage of the plant.

QTLs for FHB resistance have previously been found on all seven barley chromosomes [24, 31, 77, 81, 82]. For most of the resistance varieties, QTLs associated with FHB were detected on the long arm of chromosome 2H [30, 31, 83]. In addition, the QTLs for disease resistance and reduced DON concentration have been linked to spike morphology controlled by *vrs1* and a major HD locus (P*pd-H1*) [90]. The number of detected QTLs varies in different reports, ranging from only one in the study by Mesfin et al. [24], two [4, 31, 78], and up to to 10 [22]. For many FHB regions in the barley genome, QTLs for DON concentration have been detected for both barley [83, 91] and wheat [87, 92], although such a relationship is not reported as significant in all studies [30]. Identification of QTLs linked to FHB symptoms can be confounded by agronomic traits such as HD, plant height, and properties associated with spike morphology [24, 82]. Hence, mapping of traits characterized by strong phenotypic correlations constitutes a challenge in terms of pleiotropy/linkage. Massman et al. [90] summarized previously described FHB regions and showed all detected QTLs associated with genome location (bin). The QTLs were located on chromosome 2H at three different spots (bin 8, bin 10, and bin 13–14). In our study, six QTLs related do FHBi were found. Of these QTLs, three were identified on chromosome 2H at positions 23.1, 87.7, and 216.7 cM, corresponding to the previously mentioned bin locations. Three other loci–*QFHB.IPG-3H*, *QFHB.IPG-5H*, and *QFHB. IPG-7H* –were found on chromosomes 3H, 5H, and 7H, respectively. *QFHB.IPG-2H_1* was found on the short arm of chromosome 2H in the vicinity of SNP marker *BOPA1_5880–2547*, which explains the largest percentage of phenotyping variance (3.69–30.69) of all FHB QTLs detected. The CamB alleles positively contributed to the increase in FHBi at this locus, which is in accordance with previous studies in which early heading plants were vulnerable to FHB symptoms. In our study, the main QTL for HD was located on chromosome 2H in the vicinity of marker *BK_12* at position 22 cM, shifted 1.1 cM from marker *BOPA1_5880–2547*. According to Turner at al. [93], the most significant SNP marker (*BK_12*) is directly located within the *Ppd-H1* gene, which is the main determinant of response to long day conditions in barley. The 2Hb8 QTL is also considered to be a major locus for resistance to FHB and DON accumulation [94]. Delayed head emergence may increase the likelihood that the host will escape infection by the pathogen [76, 95]. On the other hand, late heading is undesirable in breeding programs addressed to arid regions [96]. Plants with lower FHB severities usually have one or more of the following traits: late heading, increased height, and two-rowed spike morphology [64, 65, 75]. Although tall plants are usually more resistant to disease than short plants [78], the heading date can be either negatively [76, 78] or positively correlated with DON content in seeds [22, 24]. The major QTL associated with heading and located on chromosome 2H (*Q. HD.LC-2H*) was also identified at SNP marker *5880–2547* in our previous study [37]. SNP *5880–2547* was the closest marker to QTLs associated with plant architecture, spike morphology, and grain yield in those experiments.

Plant height is under polygenic control and represents one of the most important agronomic traits for barley [97, 98]. The right timing of flowering time allows optimal grain development with regards to the availability of heat, light, and water, while semi-dwarf cereals allocate more resources into grain production than taller plants and show reduced losses

through lodging [99, 100]. In addition, due to increasing of moisture content of plants, lodging causes expansion of the infection [101]. In the current study, seven loci for LSt were detected. The main locus (*QLSt.IPG-2H_1*) was on chromosome 2H in the vicinity of marker *BK_13*, which coincided with the main HD QTL. In this study, only one locus was found on chromosome 3H, where *sdw1/denso* gene has been located in our previous investigations [97, 102]. There is a gradient in ascospore concentration from the soil surface to upper part of plant stem. Thus, short plants tend to have higher level of FHB infection [103], which is in accordance with our results.

In barley, spike length and spike characters such as number of grains and spikelets per spike are perceived as important agromorphological traits due a direct impact on crop yield [104]. Spike architecture has significant influence on yield and less dense spike alters the spike microenvironment by making it less favorable for fungal infection [105]. In the current study, six QTLs linked to Density were found. Of the six QTLs detected, four loci were found on chromosome 2H. The major QTL (*QDen.IPG-2H-1*) was located on the short arm of 2H in the vicinity of marker *BK_12*. Two QTLs related to the density of the spike were found on chromosome 5H. In most cases, CamB alleles contributed positively to this trait. In many studies, plants with lax spikes have been reported as being less vulnerable for fungal infection [91, 105]. On the other hand, Yoshida et al. [77] found no differences between genotypes when comparing barleys with normal and dense type of spikes. Steffenson et al. [75] showed that FHB severity was higher in dense spike NILs vs. lax spike plants, but no significant differences were found. Langevin et al. [106], in a study using barley with two- and six-row types of spikes, concluded that the high level of DON contamination observed in dense spikes occurred mainly because of direct contact with florets. To summarize, the results for an association between disease severity and spike architecture of barley plants are not consistent.

In our study FHB QTLs coincidence with traits connected with spike morphology, HD and height (LSt) on chromosomes 2H and 5H was found. The underlying mechanism of coincident HD, LSt, Density and disease QTL could be due to tight linkage or pleiotropy. However, late-heading plants may serve as an escape mechanism from infection due to a lack of overlapping periods in plant development and fungus life cycle. Plant height could contributed to physically avoiding pathogens as well as inflorescence structure [83].

The GO term overrepresentation analysis combines information from regions containing QTLs for a given trait and gene function terms. Thus, we investigated the GO term over-representation of three hotspots containing, among others, QTLs for FHB. Overrepresentation analysis revealed GO annotations linked to glycosylation process. Two annotations were assigned to region B (with GO-ID: 15018 and 15020). Both annotations were referred to genes associated with glycosyltransferase (HORVU2Hr1G013590 and HORVU2HHr1G013630). Another two annotations in the E region were related to genes referred to UDP-glycosyltransferase superfamily protein (HORVU5Hr1G095010, HORVU5Hr1G096240, HORVU5Hr1G096260, HORVU5Hr1G096310, HORVU5Hr1G096320, HORVU5Hr1G096340, HORVU5Hr1 G096360). Glycosylation is a widespread cellular modification reaction in all living organisms, attaching a carbohydrate to the hydroxyl or other functional group of a molecule in a biosynthetic pathway [107]. Glycosylation modifications are catalyzed by glycosyltransferase enzymes (GTs), which are highly divergent, polyphyletic and belong to a multigene family [108]. Plant uridine diphosphate (UDP)-glucosyltransferases (UGT) catalyze the glucosylation of xenobiotic, endogenous substrates and phytotoxic agents produced by pathogens such as mycotoxins [109, 11]. The studies have shown that plant UDP-glucosyltransferase genes have significant role in plant resistance both to biotic and abiotic stresses [111, 112]. Poppenberger et al. [113] demonstrated that DON resistance can be achieved by the enzymatic conversation (a natural detoxification process in plants called glycosylation) of the toxin into the non-toxic form

(DON-3-0-glucoside) by UDP-glucosyltransferase. Recently the HvUGT-10 W1 gene has been isolated from an FHB resistant barley variety conferred FHB tolerance [110]. It is also worth to mention that in our study these GO terms have been annotated for two regions, where FHBi QTLs were found on chromosomes 2H and 5H in this study.

## Supporting information

**S1 Fig. Seeds, observed in Lcam plants, with moderate or severe Fusarium symptoms.**
(DOCX)

**S2 Fig. Abundant mycelial growth observed on the grain surface.**
(DOCX)

**S3 Fig. The positions of QTLs (chromosomes 1H and 2H) detected for studied traits.**
(DOCX)

**S4 Fig. The positions of QTLs (chromosomes 3H, 4H and 5H) detected for studied traits.**
(DOCX)

**S5 Fig. The positions of QTLs (chromosomes 6H and 7H) detected for studied traits.**
(DOCX)

**S1 Table. The mean values for studied traits for parental cultivars.**
(DOCX)

**S2 Table. The mean values for studied traits for RILs.**
(DOCX)

**S3 Table. ANOVA results, variance components and heritability estimated for studied traits.**
(DOCX)

**S4 Table. QTLs identified in the LCam population for the observed traits.**
(DOCX)

**S5 Table. An overrepresentation analysis associated with three regions (hotspots B, E, and F).**
(DOCX)

## Author Contributions

**Conceptualization:** Piotr Ogrodowicz, Anetta Kuczyńska.

**Data curation:** Piotr Ogrodowicz, Anetta Kuczyńska, Krzysztof Mikołajczak.

**Formal analysis:** Piotr Ogrodowicz, Anetta Kuczyńska, Krzysztof Mikołajczak, Paweł Krajewski.

**Investigation:** Piotr Ogrodowicz, Anetta Kuczyńska, Michał Kempa.

**Methodology:** Piotr Ogrodowicz, Anetta Kuczyńska, Krzysztof Mikołajczak, Tadeusz Adamski, Maria Surma, Hanna Ćwiek-Kupczyńska, Michał Kempa, Michał Rokicki, Dorota Jasińska.

**Project administration:** Anetta Kuczyńska.

**Software:** Paweł Krajewski, Hanna Ćwiek-Kupczyńska.

**Supervision:** Anetta Kuczyńska.

**Validation:** Paweł Krajewski, Hanna Ćwiek-Kupczyńska.

**Visualization:** Paweł Krajewski, Hanna Ćwiek-Kupczyńska.

**Writing – original draft:** Piotr Ogrodowicz, Anetta Kuczyńska.

**Writing – review & editing:** Piotr Ogrodowicz, Anetta Kuczyńska, Krzysztof Mikołajczak, Tadeusz Adamski, Maria Surma, Paweł Krajewski, Hanna Ćwiek-Kupczyńska, Michał Kempa, Michał Rokicki, Dorota Jasińska.

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
