## [Decision Letter · Decision Letter 0]

19 Sep 2019

PONE-D-19-23971

Mapping of Quantitative Trait Loci for Traits linked to Fusarium Head Blight Symptoms Evaluation in Barley RILs

PLOS ONE

Dear Mrs Kuczyńska,

Thank you for submitting your manuscript to PLOS ONE. After careful consideration, we feel that the manuscript needs major revisions to  fully meet PLOS ONE’s publication criteria as it currently stands. Therefore, we invite you to submit a revised version of the manuscript that addresses the points raised during the review process. Comments from the reviewer#2 are in the attached file.

We would appreciate receiving your revised manuscript by Nov 03 2019 11:59PM. To enhance the reproducibility of your results, we recommend that if applicable you deposit your laboratory protocols in protocols.io, where a protocol can be assigned its own identifier (DOI) such that it can be cited independently in the future. For instructions see: http://journals.plos.org/plosone/s/submission-guidelines#loc-laboratory-protocols

We look forward to receiving your revised manuscript.

Kind regards,

Ajay Kumar

Academic Editor

PLOS ONE

Journal Requirements:

Additional Editor Comments (if provided):

Reviewers' comments:

Reviewer's Responses to Questions

**Comments to the Author**

1. Is the manuscript technically sound, and do the data support the conclusions?

Reviewer #1: Yes

Reviewer #2: Yes

2. Has the statistical analysis been performed appropriately and rigorously? 

Reviewer #1: Yes

Reviewer #2: Yes

3. Have the authors made all data underlying the findings in their manuscript fully available?

Reviewer #1: No

Reviewer #2: Yes

4. Is the manuscript presented in an intelligible fashion and written in standard English?

Reviewer #1: No

Reviewer #2: No

5. Review Comments to the Author

Reviewer #1: I reviewed the manuscript entitled “Mapping of quantitative trait loci for traits linked to Fusarium head blight symptoms evaluation in barley RILs” submitted by Ogrodowicz et al. The authors studied the association between a number of agronomic traits and resistance/susceptibility to Fusarium head blight (FHB) in 100 RILs from a barley bi-parental population Lubuski x Cam/B1/ CI08887///CI05761. A total of 70 QTL for agronomic and FHB resistance were mapped in this study. The authors used only 100 RILs for mapping QTL for complex traits like FHB resistance and agronomic traits which is something I do not support in this work. I recommend representing all the 70 QTL in a Figure (a map) rather than a table and add the SNP physical positions. The authors need to give more emphasis to the FHB/DON QTL that are co-localized with agronomic trait QTL and give their recommendations to breeders based on their findings. The authors need to improve the English and the discussion section of this manuscript. Please avoid repeating results in the discussion. These are additional comments to the authors.

1) The manuscript title is not good

2) Abstract: very long introduction in the abstract, summarized it and focus more on the findings of your study. Include the species name of Fusarium used for inoculation (line #24)

3) Introduction: there are some unnecessary details and some spelling mistakes

4) Materials and Methods:

a. Include why you chose these specific two parents for your study in materials and methods and be consistent with the names of the parents. The authors sometime mention the names of the parents and the origins of the parents (Syrian, European) in other cases. 100 RILs is a small number for mapping QTL for FHB resistance and agronomic traits.

b. Line # 116: I think you can use un-inoculated and inoculated plots instead of “V1-variant-control” and “V2-Variant-inoculation”

c. Change “Methodology” to something like “Inoculum preparation” and give more details on inoculum preparation. Why you did not use F. graminearum for inoculation. Why did you use F. Culmorum? Any reason? What the authors mean by micro-irrigation? Give more details.

d. Line 129: 10 randomly selected plants per plot? if so add “per plot”in Line# 30: can you define “stature” of plants and how that is different from plant height and if there is a scale, please describe it.

e. Describe all the traits you measure in the text and do not just refer the readers to table1.

f. Include in the methods when did you score for FHB severity (how many days after heading)

g. Adjust column width of table 1. I believe you mean “plot” not “pot” in table 1

h. Include more details on how you extracted DON (how much grain were used to quantify the toxin, method of DON extraction, more details on the ELISA methodology, if you included controls in your ELISA plates, were the samples duplicated or just included once, etc)

i. Fig1 and Fig 2 can go to supplementary

j. Line 176 and 177: replace “7.842 SNPs” by “7,842 SNPs” and same for “2.832”

k. Change “map construction” to something like “linkage map….”

l. Line 186: “markers with other segregation ratios were categorized as odd” what do you mean by “odd” do you mean markers with segregation distortion?

m. Line 186: Not clear what you mean by “incorrect regions of the chromosomes….”

n. Line 191: “recombination frequency was set at level <4”, it should be 0.4 or 40%

o. For “P” values. The “P” should be italics

p. Line 209: “exceed 20/15 %” is this a typing mistake?

5) Results:

a. Figure 3. Is very blurry: provide better quality figure

b. Line 240: “The parental forms were differentiated in terms of all studied characters”: This statement is not accurate because in Fig 3 there was no much difference in these traits LSt, FHBi, FDKn, FDKw, HLKn between the parents.

c. Better have the DON values in ppm

d. Line 268: FHBi was positively correlated with sterility. Please correct

e. Did you check for normality of traits before doing correlation ?

f. Why you did not do correlation of agronomic traits with DON levels

g. It will be good if you calculate the heritability of each trait

h. Table2. ANOVA should go to supplementary

i. Is table 3 for correlation between FHB severity or FHBi with other traits?

j. Table 2: I expected NSS and density to be positively correlated with FHB. How do you explain the negative correlations in your study?

k. Linkage map construction and table 4: add more statistics on the map. How many loci these 1,947 SNPs represent? How many markers have segregation distortion? Table 4 “map lenght” misspelled

l. All markers and QTL names should be italics

m. You should include the physical position of the markers linked to your QTL

n. Line 313: for the QTL “QNSS.IPG-2H_1” indicate which parent provides the resistant allele.

o. Line 340: font difference in the QTL name

p. Where are the QTL for DON?

q. For the co-localized QTL. I would like to see more emphasis on what FHB/DON QTL co-localized with QTL for spike and agronomic traits.

r. Table 4 could be better represented in a map so it will be easier to see which QTL are co-localized and put the physical positions of the markers.

s. I don’t see the meaning of looking for gene candidates within ± 2 cM of the FHB QTL. It is a very huge physical distance especially that the resolution of your map wouldn’t be good enough knowing that you used only 100 RILs for mapping the QTL.

6) Discussion

a. Authors should work better on the discussion of this manuscript and avoid repeating results in discussion.

b. Line 501-514: Lubuski is less susceptible to FHB in terms of DON but you have higher FHBi for Lubuski under inoculation: how do you explain that? Lines 501-509 are results not discussion

c. Line 526-530: what is the relationship between antibody specific mycelial proteins and DON measures with ELISA? The antibody in the ELISA are specific to DON not to the fungal mycelium. Your statement was not clear.

d. Line 540-542: you have negative correlation between density and FHBi which means compactness is negatively correlated with FHB but you are discussing that compactness is positively correlated with FHB. There is contradiction here.

e. What is the difference between your present study and your previous study: line 545-549: was it just the density of mapping by increasing the number of markers used?

f. Change “investigation” to “study”

g. Discuss the type of linkage between alleles providing resistance to FHB and the other agronomic traits in your study.

Reviewer #2: The manuscript does present interesting results related to FHB in barley. However, apart from few technical comments, the manuscript needs to be rewritten (except discussion portion) completely in an intelligible fashion and standard english communication skills.

6. PLOS authors have the option to publish the peer review history of their article (what does this mean?). If published, this will include your full peer review and any attached files.

Reviewer #1: No

Reviewer #2: Yes: Ravinder Singh, Asstt. Prof., SKUAST-Jammu, India (180009)

---

## [Author Response · Author response to Decision Letter 0]

6 Nov 2019

Reviewer Report

Manuscript # PONE-D-19-23971

Full Title: Mapping of Quantitative Trait Loci for Traits linked to Fusarium Head Blight Symptoms Evaluation in Barley RILs

General outline of the manuscript 

The manuscript titled “Mapping of Quantitative Trait Loci for Traits linked to Fusarium Head Blight Symptoms Evaluation in Barley RILs” presented results of QTL analysis conducted for fusarium head blight (FHB) in barley. The experiments were conducted at three locations (with three replicates each) for two conditions – control and inoculated conditions (taken as treatments). The genetic mapping for QTL analysis carried out using 9K Illumina infinium arrays has led to the identification of 70 QTLs for various traits related to FHB on different chromosomes of barley. 

Comments

Overall, the research has relevance to the identification of genetic loci controlling FHB in barley. The QTLs identified constitute an important resource for the improvement of FHB in barley. However, it still has enough room for improvement before making it available to public. 

Technical comments: (i) The words ‘symptoms evaluation’ and ‘RILs’ can be deleted from the title – it has been done. The new title is: Mapping of Quantitative Trait Loci for Traits linked to Fusarium Head Blight In Barley

 (ii) It is not clear from the text whether the variance components reported in Table 2 were calculated within each location or across all the locations combined. Ideally, it would make more sense if variance components for line x location, line x treatment interactions are calculated within each location. – Variance components reported in Table 2 were computed in the mixed linear model that is described in Methods and was applied to original data at replication level, without combining or averaging over any factor. Please note that variance components for interaction of lines and locations cannot be computed at each location.

(iii) In Table 3, the correlation coefficients were reported between FHB and traits recorded for the current study. Again, the data for individual location might be more informative than combining it across all location and subsequently using for correlation analysis. Moreover, the trait Stature was most likely recorded as an ordinal data and therefore cannot be used for calculation of correlation coefficients. – We computed correlations of traits within each location and treatment; they are reported in new Table 2. The corresponding fragment in Results has been changed correspondingly (lines 250-256):

The values of correlation coefficients between the studied traits and FHBi were generally low (Table 2). FHBi was negatively correlated with all studied traits (exeptions: FDKw and Sterility). In LES location was recorded positive significant correalation between FHBi and Density, whereas negative correlation coefficients were noted between these traits in other two locations. Positive correalation was recorded between DON content and FHBi in one of three locations (TUL) for both type of treatments. No significant correlations between DON content and other agronomic traits were observed.

 The authors must look into the fact that traits like NSS, NGS, Density, GWS and GY were negatively correlated with FHB; yet the correlation coefficient between FHB and TGW was found to be non-significant. Conducting location specific correlation analysis might point to something more important that was probably missed in the combined analysis.- it has been done. Thank you for this important remark. Indeed, detailed correlation analysis has provided some additional data (Table 2). 

 Through the manuscript, the authors have been non-consistent in the usage of the names of the genotypes.he two parents have been referred to by various names like Syrian breeding line has been written as Syrian genotype, Syrian parent and CamB; and similarly Polish Cultivar has been referred to as Lubuski, European parent, European genotype and European parent genotype. – it has been done.

 The methodology in material and methods needs to be explained a bit more. Is 20mts a standard isolation distance for fungal spores (Line 117)?- the description of methodology has been extended. Distance between plots with different treatments were established based on our previous pilot experiment determining the safe distance between areas subjected to inoculation and plots used for control experiments.

The figure of concentration of inoculum on Line 124 is not clear. – it has been done (line 118):

Conidia concentration was adjusted to 105/ml.

The number mentioned on Line 209 are not clear, as what do they refer to.- it has been clarified.

 The ‘LCam population’ on Line 229 is not defined in materials and methods.- The studied population has been defined (lines 96-98):

A 100-RIL population of spring barley (hereafter referred to as LCam) obtained from the cross between the Polish cultivar Lubuski and a Syrian breeding line - Cam/B1/CI08887//CI05761 (hereafter referred to as CamB) was studied in field conditions, together with both parental forms.

 Throughout the manuscript, the words ‘main QTLs’ have been used but not explained in the materials; probably, it referred to major QTLs.- it has been corrected.

 The words ‘trait’ and ‘character’ have been used interchangeably; the authors should have restricted to the use of one type of word only.- it has been done.

 The discussion part is well written (requires little changes only) and results have discussed in the right context. It has been done.

Comments related to written English: It seems the authors have paid very little attention to the use of good written communication skills. - There are numerous instances throughout the manuscript of misspelling very commonly used words like disease, classified, pollen containing, described etc (please refer to lines 21, 72, 87, 100, 305, 330, 332, 369, 383 and 388).- it has been corrected.

 Overall, the sentences have been written without focusing on the subject and objects of the sentences. The second paragraph on Page 4 needs to re-written.- it has been done.

 The sentence beginning on line 179 needs to be re-written.- it has been done.

 On Line 304, the sentence should begin with numbers in words only. – it has been done.

5. Review Comments to the Author

Reviewer #1: I reviewed the manuscript entitled “Mapping of quantitative trait loci for traits linked to Fusarium head blight symptoms evaluation in barley RILs” submitted by Ogrodowicz et al. The authors studied the association between a number of agronomic traits and resistance/susceptibility to Fusarium head blight (FHB) in 100 RILs from a barley bi-parental population Lubuski x Cam/B1/ CI08887///CI05761. A total of 70 QTL for agronomic and FHB resistance were mapped in this study. The authors used only 100 RILs for mapping QTL for complex traits like FHB resistance and agronomic traits which is something I do not support in this work. – references of research using similar number of plants have been added (lines 528-529):

Previous studies have used population sizes comparable to this study and successfully identified FHB QTL [24, 31, 82]. 

I recommend representing all the 70 QTL in a Figure (a map) rather than a table and add the SNP physical positions. – A new figure (S3 Figure) showing SNP positions in the reference genome has been made.

The authors need to give more emphasis to the FHB/DON QTL that are co-localized with agronomic trait QTL and give their recommendations to breeders based on their findings. – it has been done (lines 402-404):

All QTLs linked to FHB on chromosomes 2H and 5H co-localized with other agronomic traits. A total of eight chromosomal regions (named A-G) harboring QTLs for the studied traits were defined. These regions (hotspots), listed in Table 4.

(lines 576-580):

Delayed head emergence may increase the likelihood that the host will escape infection by the pathogen [75, 94]. On the other hand, late heading is undesirable in breeding programs addressed to arid regions [95]. Plants with lower FHB severities usually have one or more of the following traits: late heading, increased height, and two-rowed spike morphology [63, 64, 74].

The authors need to improve the English and the discussion section of this manuscript. Please avoid repeating results in the discussion - it has been done.

 These are additional comments to the authors.

1) The manuscript title is not good – the title has been changed. 

2) Abstract: very long introduction in the abstract, summarized it and focus more on the findings of your study. Include the species name of Fusarium used for inoculation (line #24) – it has been done.

3) Introduction: there are some unnecessary details and some spelling mistakes – the unnecessary details have been removed and spelling mistakes have been corrected. 

4) Materials and Methods:

a. Include why you chose these specific two parents for your study in materials and methods and be consistent with the names of the parents. The authors sometime mention the names of the parents and the origins of the parents (Syrian, European) in other cases. 100 RILs is a small number for mapping QTL for FHB resistance and agronomic traits. –

b. Line # 116: I think you can use un-inoculated and inoculated plots instead of “V1-variant-control” and “V2-Variant-inoculation”- it has been done.

c. Change “Methodology” to something like “Inoculum preparation” and give more details on inoculum preparation. Why you did not use F. graminearum for inoculation. Why did you use F. Culmorum? Any reason? What the authors mean by micro-irrigation? Give more details. – more details about inoculum preparation have been added and the explanation of F. culmorum using has been added to Discussion (lines 451-454):

FHB, caused by Fusarium culmorum, is a very important disease affecting crops on a global scale [9]. The pathogen is dominant in cooler areas like north, central and western Europe [54]. F. graminearum predominates in the warmer, humid areas of the world such as USA [55, 56].

d. Line 129: 10 randomly selected plants per plot? if so add “per plot”in Line# 30: can you define “stature” of plants and how that is different from plant height and if there is a scale, please describe it.- it has been done and the analysis of trait Stature has been removed from the paper.

e. Describe all the traits you measure in the text and do not just refer the readers to table1.- the traits have been divided into subgroups and brief descriptions have been given in methodology. 

f. Include in the methods when did you score for FHB severity (how many days after heading)-

it has been done (line 143):

The assessments were performed 20 days after inoculation.

g. Adjust column width of table 1. I believe you mean “plot” not “pot” in table 1- it has been done.

h. Include more details on how you extracted DON (how much grain were used to quantify the toxin, method of DON extraction, more details on the ELISA methodology, if you included controls in your ELISA plates, were the samples duplicated or just included once, etc) – 

it has been done (lines 114-122):

Fusarium culmorum isolates were incubated on wheat grain (50 g) in 300 ml Erlenmeyer glass flasks for 5 weeks. The colonies were covered with 15 ml of sterile distilled water and autoclaved twice for 30 min within 2 days. Inoculum was prepared just before the inoculations by liquid cultures of Fusarium culmorum (isolate KF846) and 0.0125% TWEEN®20 (Sigma-Aldrich Chemie GmbH). Conidia concentration was adjusted to 105/ml. Inoculation was performed at the flowering stage (Zadoks scale 65). Mist irrigation to promote fungal infection was performed for three days in the field using a sprinkler system with DN881A-type sprinkler heads equipped with 1.50-mm-diameter nozzles (Sun Hope Inc., Meguro-ku). Water was applied three times daily (at 07.00, 13.00, and 19.00) for 15 min at each interval.

i. Fig1 and Fig 2 can go to supplementary- it has been done.

j. Line 176 and 177: replace “7.842 SNPs” by “7,842 SNPs” and same for “2.832”- it has been done.

k. Change “map construction” to something like “linkage map….”- it has been done.

l. Line 186: “markers with other segregation ratios were categorized as odd” what do you mean by “odd” do you mean markers with segregation distortion?- Yes, the sentence has been corrected. 

m. Line 186: Not clear what you mean by “incorrect regions of the chromosomes….”- The statement was confusing and has been removed form manuscript.

n. Line 191: “recombination frequency was set at level <4”, it should be 0.4 or 40%- it has been done.

o. For “P” values. The “P” should be italics- it has been done.

p. Line 209: “exceed 20/15 %” is this a typing mistake?- Yes, the sentence has been corrected.

5) Results:

a. Figure 3. Is very blurry: provide better quality figure- it has been done. The new figure 1 has been added in better quality.

b. Line 240: “The parental forms were differentiated in terms of all studied characters”: This statement is not accurate because in Fig 3 there was no much difference in these traits LSt, FHBi, FDKn, FDKw, HLKn between the parents.- This sentence has been removed.

c. Better have the DON values in ppm- it has been done.

d. Line 268: FHBi was positively correlated with sterility. Please correct- it has been corrected.

e. Did you check for normality of traits before doing correlation ?- We did not perform a formal normality test of the observations. However, Figure 1 indicates that the distributions in most cases are not very nonsymmetric or far from normal. So, we think that the statistical test that were used in ANOVA and correlation analysis are justified.

f. Why you did not do correlation of agronomic traits with DON levels- no correlations between studied traits and DON content have been observed in our study. This has been reported in Results (lines 255-256):

No significant correlations between DON content and other agronomic traits were observed.

g. It will be good if you calculate the heritability of each trait- Heritabilities has been computed, separately for control and infected variants, over locations, in an appropriate linear mixed model. They are reported in S3 Table.

h. Table2. ANOVA should go to supplementary- it has been done.

i. Is table 3 for correlation between FHB severity or FHBi with other traits?- yes, it has been corrected.

j. Table 2: I expected NSS and density to be positively correlated with FHB. How do you explain the negative correlations in your study?- the association between two traits strongly linked to NSS (Density and Sterility) and FHB has been explained (lines 599-610):

Spike architecture has significant influence on yield and might alter the spike microenvironment by making it less favorable for fungal infection [104]. In the current study, six QTLs linked to Density were found. Of the six QTLs detected, four loci were found on chromosome 2H. The major QTL (QDen.IPG-2H-1) was located on the short arm of 2H in the vicinity of marker BK_12. Two QTLs related to the density of the spike were found on chromosome 5H. In most cases, CamB alleles contributed positively to this trait. In many studies, plants with lax spikes have been reported as being less vulnerable for fungal infection [90, 104]. On the other hand, Yoshida et al. [76] found no differences between genotypes when comparing barleys with normal and dense type of spikes. Steffenson et al. [74] showed that FHB severity was higher in dense spike NILs vs. lax spike plants, but no significant differences were found. Langevin et al. [105], in a study using barley with two- and six-row types of spikes, concluded that the high level of DON contamination observed in dense spikes occurred mainly because of direct contact with florets.

k. Linkage map construction and table 4: add more statistics on the map. How many loci these 1,947 SNPs represent? How many markers have segregation distortion? Table 4 “map lenght” misspelled- all possible information has been included in “Materials and methods”

l. All markers and QTL names should be italics- it has been done

m. You should include the physical position of the markers linked to your QTL – it has been included

n. Line 313: for the QTL “QNSS.IPG-2H_1” indicate which parent provides the resistant allele.- it has been done (lines 288-289):

In this case, significant QTL × E interaction was noted and Lubuski alleles conferred a positive effect in increasing this trait.

o. Line 340: font difference in the QTL name- it has been corrected.

p. Where are the QTL for DON?-

No QTLs for DON content were found in our study and the comments have been given in Discussion (lines 532-549):

In our study different tools for FHB evaluation have been used: among others: DON content estimation. No QTLs for DON content were detected but visual assessment of FHB severity like FHBi, FDK and HLK were employed here for evaluation of the level of FHB severity. In this study six, four and five QTLs were found for FHBi, FDK and HLK, respectively. The association between Fusarium head blight (FHB) intensity and DON accumulation in harvested grain is not fully understood. Varying degrees of association between Fusarium head blight intensity and DON accumulation in harvested grain have been reported in the literature, including situations with high positive correlations, low significant correlations, and negative correlations, as well as correlations close to zero [83–86]. Visual assessments of disease were usually made at Feekes GS 11.2, based on the proportion of the spike diseased, while DON was quantified in this study after harvest as the amount of DON per unit weight of a bulked sample of ground kernel. The measurement of DON in an assay typically is a composite value for seeds with different levels of DON (including those with 0 ppm) and different levels of fungal colonization. In our study positive correlation between DON content and FHBi was observed only in one location which can be explained by the fact that the growth of the fungus and the production of DON are highly weather dependent [87, 88]. Moreover, DON concentration may have increased at differential rates in the different studies, affecting the relationship between DON sampled at harvest and disease assessed different developmental stage of the plant. 

q. For the co-localized QTL. I would like to see more emphasis on what FHB/DON QTL co-localized with QTL for spike and agronomic traits.- it has been done (lines 613-618):

In our study FHB QTLs coincidence with traits connected with spike morphology, HD and height (LSt) on chromosomes 2H and 5H was found. The underlying mechanism of coincident HD, LSt, Density and disease QTL could be due to tight linkage or pleiotropy. However, late-heading plants may serve as an escape mechanism from infection due to a lack of overlapping periods in plant development and fungus life cycle. Plant height could contributed to physically avoiding pathogens as well as inflorescence structure [82].

r. Table 4 could be better represented in a map so it will be easier to see which QTL are co-localized and put the physical positions of the markers.- A new figure (S3 Figure) has been made

s. I don’t see the meaning of looking for gene candidates within ± 2 cM of the FHB QTL. It is a very huge physical distance especially that the resolution of your map wouldn’t be good enough knowing that you used only 100 RILs for mapping the QTL.- 

This section in Discussion has been changed. Genomic and functional genomic studies normally generate large lists of interesting genes, and translating such lists into biologically meaningful information is critical to understand the underlying regulatory mechanisms of the related biological processes. That’s why the overrepresentation analysis of gene functions based on Gene Ontology term was employed in our study. The basis assumption underlying the overrepresentation approach is that in several of multiple QTL regions for a given trait, a causal gene is present, and that several causal genes have related or similar gene functions (Beissbarth and Speed, 2004). We believe this type of analysis will be useful for looking up a cues about the potential factors contributing to traits importance.

Ref: Beissbarth T, Speed TP. GOstat: find statistically overrepresented Gene Ontologies within a group of genes. Bioinformatics. 2004;20:1464–1465.

6) Discussion

a. Authors should work better on the discussion of this manuscript and avoid repeating results in discussion.- it has been done.

b. Line 501-514: Lubuski is less susceptible to FHB in terms of DON but you have higher FHBi for Lubuski under inoculation: how do you explain that? Lines 501-509 are results not discussion- it has been done (lines 469-470):

This can be explained by the fact that symptomless grains may contain significant amounts of mycotoxins, while symptomatic grains within the same samples may not [61].

c. Line 526-530: what is the relationship between antibody specific mycelial proteins and DON measures with ELISA? The antibody in the ELISA are specific to DON not to the fungal mycelium. Your statement was not clear.- The statement was confusing. It has been removed from paper and the explanation of low correlations between FHB and DON content has been added.

d. Line 540-542: you have negative correlation between density and FHBi which means compactness is negatively correlated with FHB but you are discussing that compactness is positively correlated with FHB. There is contradiction here.- it has been corrected.

e. What is the difference between your present study and your previous study: line 545-549: was it just the density of mapping by increasing the number of markers used?- the density of mapping is the main difference between those maps but the previous statement has been removed form manuscript because of its unimportance for discussion.

f. Change “investigation” to “study”- it has been done.

g. Discuss the type of linkage between alleles providing resistance to FHB and the other agronomic traits in your study.- 

results and discussion have been changed:

QHLKw.IPG-2H was found in the vicinity to marker BK_12. In the same position a set of QTLs linked to different agronomic traits was found (Density, GWS, GY, HD, NGS and NSS - QDen.IPG-2H_1, QGWS.IPG-2H_1, QGY.IPG-2H, QHD.IPG-2H, QNGS.IPG-2H and QNSS.IPG-2H_1, respectively). QHLKn.IPG-2H_2, other QTL related to FHB, was detected in the vicinity of marker BK_13 – in the same position as QTLs related to LS (QLS.IPG-2H) and LSt (QLSt.IPG-2H_1). In both cases Lubuski contributed positively to the increase of the trait linked to HLK (HLKn and HLKw).

In our study FHB QTLs coincidence with traits connected with spike morphology, HD and height (LSt) on chromosomes 2H and 5H was found. The underlying mechanism of coincident HD, LSt, Density and disease QTL could be due to tight linkage or pleiotropy.

Reviewer #2: The manuscript does present interesting results related to FHB in barley. However, apart from few technical comments, the manuscript needs to be rewritten (except discussion portion) completely in an intelligible fashion and standard english communication skills.- manuscript has been rewritten. The English language paper was revised by mother tongue.

---

## [Decision Letter · Decision Letter 1]

6 Dec 2019

PONE-D-19-23971R1

Mapping of Quantitative Trait Loci for Traits linked to Fusarium Head Blight In Barley

PLOS ONE

Dear Mrs Kuczyńska,

Thank you for submitting your manuscript to PLOS ONE. After careful consideration, we feel that it has merit but still needs some minor revision. Therefore, we invite you to submit a revised version of the manuscript that addresses the points raised during the review process.

We would appreciate receiving your revised manuscript by Jan 20 2020 11:59PM. To enhance the reproducibility of your results, we recommend that if applicable you deposit your laboratory protocols in protocols.io, where a protocol can be assigned its own identifier (DOI) such that it can be cited independently in the future. For instructions see: http://journals.plos.org/plosone/s/submission-guidelines#loc-laboratory-protocols

We look forward to receiving your revised manuscript.

Kind regards,

Ajay Kumar

Academic Editor

PLOS ONE

Reviewers' comments:

Reviewer's Responses to Questions

**Comments to the Author**

1. If the authors have adequately addressed your comments raised in a previous round of review and you feel that this manuscript is now acceptable for publication, you may indicate that here to bypass the “Comments to the Author” section, enter your conflict of interest statement in the “Confidential to Editor” section, and submit your "Accept" recommendation.

Reviewer #1: All comments have been addressed

2. Is the manuscript technically sound, and do the data support the conclusions?

Reviewer #1: Partly

3. Has the statistical analysis been performed appropriately and rigorously? 

Reviewer #1: I Don't Know

4. Have the authors made all data underlying the findings in their manuscript fully available?

Reviewer #1: No

5. Is the manuscript presented in an intelligible fashion and written in standard English?

Reviewer #1: No

6. Review Comments to the Author

Reviewer #1: The authors improved the quality of their manuscripts and did most of the modifications I requested in my first revision. However the authors still need to improve the writing of this manuscript.

Additional comments

1- “parental genotypes were 107 chosen on the basis of earlier studies conducted by Górny and co-workers (lit)” put the reference here instead of “(li)”

1- “Fusarium culmorum isolates were incubated on wheat grain (50 g) in 300 ml Erlenmeyer glass flasks for 5 weeks. The colonies were covered with 15 ml of sterile distilled water and autoclaved twice for 30 min within 2 days”.

Is the water that was autoclaved or the mixture of fungal conidia suspended in H2O? I believe you don’t want to autoclave your inoculum so please rewrite this sentence. “5 weeks” and “2 days” should be “five weeks” and “two days”.

2- “Spike architecture has significant influence on yield and might alter the spike microenvironment by making it less favorable for fungal infection [104]”:

What spike architecture is influencing yield and in what direction (positive or negative?) and what spike architecture is influencing microenvironment. I believe you mean less dense spike make the microenvironment less favorable for fungal infection.

3- “In many studies, plants with lax spikes have been reported as being less vulnerable for fungal infection [90, 104]. On the other hand, Yoshida et al. [76] found no differences between genotypes when comparing barleys with normal and dense type of spikes. Steffenson et al. [74] showed that FHB severity was higher in dense spike NILs vs. ……………...”

So your data opposes all of these previous studies? (check the negative correlations between FHB and density and NSS in Table2, does your data mean that more dense spikes are less susceptible to FHB?).

4- Add number of loci per chromosome (markers mapped in the same location represent a single locus) and add percentage of distorted markers per chromosome in Table4

5- I suggest presenting table 2 as figure (correlation plots) if possible

6- “Conidia concentration was adjusted to 105/ml”. Change this to “inoculum concentration was adjusted to 105 spore/ml”

7- Table1 clolum1 and column2: please adjust the width because there some missing words.

8- Looking for candidates genes based on QTL mapping is not adequate at this stage because of the low resolution of your QTL mapping. Looking for candidate genes is appropriate only after fine mapping.

9- the authors need to provide the phenotypic and genotypic data of the this population

7. PLOS authors have the option to publish the peer review history of their article (what does this mean?). If published, this will include your full peer review and any attached files.

Reviewer #1: No

---

## [Author Response · Author response to Decision Letter 1]

16 Jan 2020

Dear Editor,

We would like to thank you for all suggestions and comments, which have been useful for improving the quality of the manuscript. All objections were properly faced and a suitable answer/modification was provided. 

Reviewer #1: The authors improved the quality of their manuscripts and did most of the modifications I requested in my first revision. However the authors still need to improve the writing of this manuscript.

Additional comments

1- “parental genotypes were 107 chosen on the basis of earlier studies conducted by Górny and co-workers (lit)” put the reference here instead of “(li)”

Thank you for the suggestion. It has been done (lines 99-100):

“The plant materials were described in detail in Ogrodowicz et al. [37] and parental genotypes were chosen on the basis of earlier studies conducted by Górny et al. [38].”

1- “Fusarium culmorum isolates were incubated on wheat grain (50 g) in 300 ml Erlenmeyer glass flasks for 5 weeks. The colonies were covered with 15 ml of sterile distilled water and autoclaved twice for 30 min within 2 days”.

Is the water that was autoclaved or the mixture of fungal conidia suspended in H2O? I believe you don’t want to autoclave your inoculum so please rewrite this sentence. “5 weeks” and “2 days” should be “five weeks” and “two days”.

It has been corrected. The sentence about sterilisation of water has been removed (insignificant detail).

2- “Spike architecture has significant influence on yield and might alter the spike microenvironment by making it less favorable for fungal infection [104]”:

What spike architecture is influencing yield and in what direction (positive or negative?) and what spike architecture is influencing microenvironment. I believe you mean less dense spike make the microenvironment less favorable for fungal infection.

Thanks for the suggestion. It has been corrected:(601)

“Spike architecture has significant influence on yield and less dense spike might alters the spike microenvironment by making it less favorable for fungal infection [104].”

3- “In many studies, plants with lax spikes have been reported as being less vulnerable for fungal infection [90, 104]. On the other hand, Yoshida et al. [76] found no differences between genotypes when comparing barleys with normal and dense type of spikes. Steffenson et al. [74] showed that FHB severity was higher in dense spike NILs vs. ……………...”

So your data opposes all of these previous studies? (check the negative correlations between FHB and density and NSS in Table2, does your data mean that more dense spikes are less susceptible to FHB?).

4- Add number of loci per chromosome (markers mapped in the same location represent a single locus) and add percentage of distorted markers per chromosome in Table4.

It has been done.

5- I suggest presenting table 2 as figure (correlation plots) if possible. 

Thank you for suggestion. It has been done.

6- “Conidia concentration was adjusted to 105/ml”. Change this to “inoculum concentration was adjusted to 105 spore/ml”

It has been changed.

7- Table1 clolum1 and column2: please adjust the width because there some missing words.

It has been done.

8- Looking for candidates genes based on QTL mapping is not adequate at this stage because of the low resolution of your QTL mapping. Looking for candidate genes is appropriate only after fine mapping.

The candidate gene strategy has shown promise for bridging the gap between quantitative genetic and molecular genetic approaches to study complex traits (Ingvarsson and Street 2011). A GO annotation is regarded as a statement about gene function of a particular gene. That’s why we changed “potential candidate genes” term on “ (gene annotations linked to potential candidate genes located in the vicinity (intervals around markers extended by ±2cM) of the particularly robust QTL.) (207-210).

Low resolution of the estimated chromosomal location of quantitative trait loci (QTL) is a major obstacle in application of QTL linkage mapping results for. Up to a certain point, mapping resolution can be improved by increasing marker density (Darvasi et al. 1993). However, for given sample size and standardized QTL substitution effect, ultimate map resolution is fixed and cannot be improved even with infinite marker density (Darvasi et al. 1993; Ronin et al. 2003). In our study, genetic map spanned 1678 cM and contained 1947 single nucleotide polymorphism markers. We believe this resolution is sufficient for a Gene Ontology analysis. Similar mapping resolution was used i.a. in study concerning wheat 2D chromosome (Deng et al. 2019). Effective sample size can also be increased by accumulating recombinants in advanced generations (Darvasi and Soller 1995). That`s why in our study a set of recombinant inbred lines (F10) was used. 

Ref:

Ingvarsson, P. K. and Street, N. R. (2011) Association genetics of complex traits in plants, New Phytologist,189(4), 909-922

A. Darvasi, A. Weinreb, V. Minke, J. I. Weller, and M. Soller (1993). Detecting marker-QTL linkage and estimating QTL gene efect and map location using a saturated geneticmap.Genetics,134, 943±951

Ronin Y.I., Korol A.B., Shtemberg M., Nevo E. & Soller M.(2003) High resolution mapping of quantitative trait lociby selective recombinant genotyping.Genetics164, 1657–66

Darvasi, A. and Soller, M. 1995. Advanced intercross lines: An experimental population for fine genetic mapping. Genetics 141: 1199-1207

Deng M, Wu F, Zhou W, Li J, Shi H, Wang Z, Lin Y, Yang X, Wei Y, Zheng Y Liu Y (2019) Mapping of QTL for total spikelet number per spike on chromosome 2D in wheat using a high-density genetic map. Genet Mol Biol. Jul-Sep;42(3):603-610. doi: 10.1590/1678-4685-GMB-2018-0122. Epub 2019 Nov 14.

9- the authors need to provide the phenotypic and genotypic data of the this population.

The phenotypic and genotypic data have been provided (line 220):

Raw data are available at [www.polapgen.pl/eksplan/dataset_FHB_in_LCamRIL.zip].

---

## [Editor Report · Decision Letter 2]

21 Jan 2020

Mapping of Quantitative Trait Loci for Traits linked to Fusarium Head Blight In Barley

PONE-D-19-23971R2

Dear Dr. Kuczyńska,

We are pleased to inform you that your manuscript has been judged scientifically suitable for publication and will be formally accepted for publication once it complies with all outstanding technical requirements.

With kind regards,

Ajay Kumar

Academic Editor

PLOS ONE
---

## [Editor Report · Acceptance letter]

28 Jan 2020

PONE-D-19-23971R2 

Mapping of Quantitative Trait Loci for Traits linked to Fusarium Head Blight In Barley 

Dear Dr. Kuczyńska:

I am pleased to inform you that your manuscript has been deemed suitable for publication in PLOS ONE. Congratulations! Your manuscript is now with our production department. 

With kind regards,

on behalf of

Dr. Ajay Kumar 

Academic Editor

PLOS ONE